# Body Temperature Drop as a Humane Endpoint in Snake Venom-Lethality Neutralization Tests

**DOI:** 10.3390/toxins15090525

**Published:** 2023-08-26

**Authors:** Rosa De Jesus, Adam E. Tratner, Alanna Madrid, Andrés Rivera-Mondragón, Goy E. Navas, Ricardo Lleonart, Gabrielle B. Britton, Patricia L. Fernández

**Affiliations:** 1Bioterio, Instituto de Investigaciones Científicas y Servicios de Alta Tecnología (INDICASAT AIP), City of Knowledge, Panama City 0843-01103, Panama; rdejesus@indicasat.org.pa (R.D.J.); amadrid@indicasat.org.pa (A.M.); 2Florida State University, Republic of Panama Campus, City of Knowledge, Panama City 0843-01103, Panama; atratner@fsu.edu; 3Centro de Neurociencias, INDICASAT AIP, City of Knowledge, Panama City 0843-01103, Panama; 4Instituto Especializado de Análisis (IEA), Universidad de Panamá, Panama City P.O. Box 3366, Panama; andres.riveraa@up.ac.pa (A.R.-M.); goy.navast@up.ac.pa (G.E.N.); 5Centro de Biología Celular y Molecular de Enfermedades, INDICASAT AIP, City of Knowledge, Panama City 0843-01103, Panama; rlleonart@indicasat.org.pa

**Keywords:** venom, antivenom, temperature, refinement

## Abstract

Snake venom neutralization potency tests are required for quality control assessment by manufacturers and regulatory authorities. These assays require the use of large numbers of mice that manifest severe signs associated with pain and distress and long periods of suffering. Despite this, many animals make a full recovery; therefore, the observation of clinical signs as a predictor of animal death is highly subjective and could affect the accuracy of the results. The use of a more objective parameter such as body temperature measurement could help establish a humane endpoint that would contribute to significantly reducing the suffering of large numbers of animals. We determined the temperature drop in BALB/c mice exposed to the mixtures of *Bothrops asper* or *Lachesis stenophrys* venom and a polyvalent antivenom by using an infrared thermometer. Our data show that, based on the temperature change from baseline, it is possible to predict which animals will survive during the first 3 h after inoculation. The data provided in this study may contribute to future reductions in animal suffering, in concordance with general trends in the use of laboratory animals for the quality control of biologicals.

## 1. Introduction

Snakebite envenoming is a neglected and life-threatening disease that constitutes a public health problem worldwide [1,2]. Rural populations are often the most affected by this disease with high rates of morbidity and mortality [3]. Clinical manifestations associated with envenoming vary depending on the snake species and can include local tissue damage, bleeding, neurotoxicity, hypovolaemic shock, cardiotoxicity, thrombosis, and so on [4,5]. Snake venoms are complex mixtures of proteins whose toxic effects are produced via their action as independent components or by forming complexes [6,7,8]. The large diversity of snake venoms and the complexity of their compositions [7] are challenges for the designing and manufacturing of new therapeutics.

The only available specific treatment for snakebite envenoming is with the use of antivenoms, which are immunoglobulins obtained from the plasma of animals, such as horses, sheep, or camels, that are hyper-immunized with venoms of one or several species of snakes [9]. In some cases, antivenoms are constituted by immunoglobulin fragments obtained via enzymatic digestion of animal plasma, reducing some adverse effects during human treatment [10]. Once the antivenom is administered to patients, the immunoglobulins or their fragments neutralize the venom toxins, reducing their deleterious effects. The dosage of antivenom is usually estimated by using neutralizing potency tests in rodents.

Evaluation of the antivenom potency is performed as part of the preclinical studies for new antivenom, and it is also required for quality-control purposes. These tests are assessed by analyzing the neutralization of venom-induced lethality in mice. The neutralization capacity or potency of the antivenom is expressed as the median effective dose (ED_50_), which means the volume of antivenom that rescues 50% of the inoculated mice from venom-induced lethality [9]. Since venom-neutralizing potency tests are performed on animals, it is essential to follow ethical conducts and guidelines that guarantee minimum animal suffering. Alternative assays have been suggested [11], but currently, regulations for the implementation of therapeutics in humans require in vivo preclinical tests [9,12]. In antivenom potency tests, animals are injected with high doses of venom and increasing doses of antivenoms. Animals exhibit clinical manifestations associated with pain and the detrimental effects of venom, ultimately dying. Lethality reduction occurs as antivenom concentrations increase [12]. Much concern has been expressed about the suffering of animals during these procedures, and several actions have been proposed to mitigate it, including: (i) reducing the number of animals used in the assays; (ii) reducing the time of the lethality assessment; (iii) the use of analgesia; and (iv) the application of a humane endpoint [9,13].

Variation in body temperature in rodents has been suggested as a criterion for establishing a humane endpoint in a variety of models [14,15,16,17]. A drop in body temperature is an indicator of disease or toxemia in animals that has been correlated with death in infectious diseases models [18,19]. A drop in body temperature has been previously reported in animals exposed to snake venom and other toxins [18,20,21,22]. However, only few studies have tested this parameter as a potential predictor of death induced via venom inoculation [18]. With the purpose of future implementation of the concept of refinement in quality control antivenom evaluations, we assessed the body temperature of animals during venom-neutralizing potency tests to determine if this parameter could be used as a predictor of mortality. BALB/c mice were exposed to venom from *Bothrops asper* or *Lachesis stenophrys* mixed with polyvalent antivenom. Our results suggest that it is possible to predict the animal death in the first 3 h after inoculation by using temperature variation as a criterion.

## 2. Results

A constant dose of *B. asper* or *L. stenophrys* venom combined or not with variable doses of antivenom were administered to mice (four groups of five animals) via intraperitoneal inoculation. The body temperature was measured after 0, 1, 2, and 3 h, and lethality was recorded for 48 h.

Descriptive analyses were used to examine the mortality rates and changes in temperature after the subjects were challenged with different venom types and antivenom serum dosages. Most subjects (53.9%) were deceased before the end of the study, and lethality rates were approximately equal for subjects exposed to *B. asper* venom (53.8%) and *L. stenophrys* venom (54%). Figure 1 shows the Kaplan–Meier curves for each type of venom–antivenom mixture. All subjects that received venom without serum or 6.75 mg venom/mL antivenom died before the end of the study, followed by 67.6% of subjects that received 4.5 mg venom/mL antivenom and 5.7% of subjects that received 3 mg of venom/mL antivenom and 2 mg of venom/mL antivenom, respectively. There were no significant differences in the baseline body temperature (*M* = 31.95, *SD* = 1.19) across groups (*p* = 0.603), and no significant differences in baseline body temperature between surviving and non-surviving subjects (*p* = 0.827), regardless of venom type. The average body temperature across groups decreased after 1 h (*M* = 25.73, *SD* = 3.03) and remained lower than the baseline body temperature for the remainder of the study (Appendix A). The lowest recorded body temperature was 19.4 °C, and the lowest recorded body temperature for a surviving subject was 21.9 °C.

### 2.1. Effect of Antivenom Dosage and Venom Type on Survival Duration

A 2 × 5 (venom type × antivenom dosage) univariate analysis of variance controlling for the effects of the baseline temperature assessed the group differences in survival duration. Tests between subjects yielded a significant main effect of dosage [*F*(4, 147) = 157.66, *p* < 0.001, *ηp*^2^ = 0.82] and a two-way interaction between antivenom dosage and venom type on survival duration [*F*(4, 147) = 3.34, *p* = 0.012, and *ηp*^2^ = 0.09]. Non-surviving subjects in the group of 6.75 mg venom/mL antivenom died sooner (all *p*-values < 0.001). Specifically, subjects that received venom without antivenom (*M* = 1.55, *SD* = 0.57) had, on average, the shortest survival duration, followed by subjects that received 6.75 mg venom/mL antivenom (*M* = 4.24, *SD* = 0.69), 4.5 mg venom/mL antivenom (*M* = 4.63, *SD* = 1.13), 3 mg of venom/mL antivenom (*M* = 5.90, *SD* = 0.40), and 2 mg of venom/mL antivenom (*M* = 5.80, *SD* = 0.92) (Table 1). No difference was found in survival duration between subjects that received 2 mg of venom/mL antivenom and 3 mg of venom/mL antivenom (*p* = 0.592). The results also indicated that subjects that received *B. asper* venom with no antivenom died 0.92 h sooner than subjects that received *L. stenophrys* venom with no antivenom (*p* < 0.001), though there were no other significant differences in survival duration between the other groups when comparing *B. asper* and *L. stenophrys* venoms at all examined group dosages (Figure 1).

### 2.2. Temperature Change from Baseline and Venom Type as Predictors of Mortality

A hierarchical binary logistic regression model, including the baseline temperature, venom type, temperature change, and antivenom dosage, was used to examine the predictors of mortality. Predictor variables were entered into the model stepwise (Table 2). The omnibus test of Step 1 was not significant (χ^2^ = 0.29, *p* = 0.589). The baseline temperature did not significantly predict mortality (*b* = −0.08, *p* = 0.589) and the overall classification accuracy of the model was poor (49.2%). Step 2 slightly improved the classification accuracy of the model (57.6%), but the omnibus test was not significant (χ^2^ = 0.93, *p* = 0.335) and the venom type did not significantly predict mortality (*b* = 0.34, *p* = 0.336). Step 3 yielded a significant omnibus test (χ^2^ = 98.70, *p* < 0.001) and greatly improved the classification accuracy (86.4%). Controlling for the venom type and the baseline temperature, the temperature change significantly predicted the mortality (*b* = 0.94, *p* < 0.001) such that subjects were 2.58 times more likely to die with each unit change in temperature. The omnibus test of Step 4 was significant (χ^2^ = 39, *p* < 0.001) and showed further improvement in classification accuracy (94.7%). Holding the constant temperature change, venom type, and baseline temperature, and venom/antivenom dosage significantly predicted the mortality (*b* = 2.78, *p* < 0.001). For each unit increase in venom/antivenom dosage, subjects were 16.17 times more likely to die for both venom types.

### 2.3. Body Temperature in Surviving vs. Non-Surviving Animals

Repeated measures ANOVA was used to assess the body temperature at baseline, 1 h, 2 h, and 3 h in subsets of animals that subsequently survived (*n* = 46) and those that did not (*n* = 31). Across venom types, ANOVA revealed a statistically significant difference in temperatures between animals that survived and those that did not [*F*(1, 75) = 120.70, *p* < 0.001, and *ηp*^2^ = 0.62]. Pairwise comparisons indicated significantly lower temperatures at all time points (except baseline) in animals that did not survive (all *p*-values < 0.001). Figure 2 shows that the pattern of temperature change across time points in surviving and non-surviving animals was similar for both venom types. Namely, significant differences between surviving and non-surviving animals were evident at each time point. Similar results were obtained when the analysis was performed in independent experiments, with significant differences at all time points between the animals that survived and those that did not (Appendix A). Only in experiment 2 with *B. asper*/antivenom treatment, we did not find differences at the 1 h time point (Appendix A), which we attributed to the reduced number of animals per group (five mice/group). For animals that survived, at 1 h, the average temperature drop was 3.9 °C (95% CI: 3.1–4.6). At this time point, the animals that would not survive had a temperature variation relative to a baseline of 6.9 °C (95% CI: 6.0–7.8). These results suggest that, for this experimental set up of using the venoms of *B. asper* and *L. stenophrys*, animals that reach a drop in temperature equal to or greater than 6 °C will not survive (Table 3). Importantly, these results also suggest that, on the basis of temperature change from baseline, it is possible to predict which animals will and will not survive within 3 h post-inoculation.

To determine which time point (1, 2, or 3 h) would be most appropriate to eventually establishing the humane endpoint, we calculated hypothetical ED_50_ values for each time point, selecting those animals that had a drop in temperature of at least 6 °C as those that would be euthanized (Appendix A). These values were compared with the ED_50_ values based on the mortalities at 48 h (Appendix A), according to the recommendations of WHO guidelines [9]. ED_50_ values were calculated via the Trimmed Spearman–Karber Method [23]. Considering that the experimental design uses the minimum number of animals recommended for antivenom potency tests in quality control evaluations (five animals/group), in some experiments, there were no partial mortalities recorded; thus, the Spearman–Karber Method did not yield confidence interval values (Appendix A).

In the experiments for the *B. asper* venom, the hypothetical ED_50_ values for the 3 h time point, but not for 1 and 2 h, were consistently similar across the experiments to those calculated according to the mortalities evaluated at 48 h (Appendix A). Although at the 1 and 2 h time points it is possible to statistically predict which animals will not survive, for ED_50_ calculation purposes, these time points might not be appropriate since the temperature drop alone could overestimate the number of animals reaching the temperature criteria to be euthanized. This situation may produce an underestimation of the true ED_50_. In the *L. stenophrys* experiments, it was only possible to calculate the ED_50_ at 3 h for experiment 1. This value is also in agreement with the ED_50_ obtained from the 48 h mortality experiments (Appendix A). For this venom, more experiments are necessary to determine the behavior of temperature at the 3 h time point. Hypothetical ED_50_ values for the 2 h time point were similar throughout the experiments to the ED_50_ calculated according to the recorded mortalities at 48 h. Altogether these results suggest that the 3 h time point could be used as a humane endpoint for BALB/c mice subjected to lethality neutralization experiments for these venoms.

## 3. Discussion

The procedures related to the manufacture and quality control of antivenoms for the treatment of snakebite envenoming require the use of animals. Rodents are commonly used to evaluate the effectiveness of antivenoms. During those procedures, animals suffer pain and distress related to the toxic effects of the venom. Therefore, the application of the 3R principles (Replacement, Reduction, and Refinement) is of the utmost urgency. To generate data that contributes to the implementation of humane endpoints to reduce animal suffering in lethality neutralization tests, we evaluated the temperature variation in mice inoculated with mixtures of *B. asper* or *L. stenophrys* venoms and antivenom. We observed that when using temperature change from baseline as a criterion, it is possible to predict, at very early treatment time points, which animals will and will not survive.

We observed that mice exposed to the toxic effects of venom from *B. asper* or *L. stenophrys* showed a sudden drop in body temperature. This decrease in temperature became more marked in experimental groups where the ratio venom/antivenom increased. As the amount of antivenom supplied decreased, the temperature changes over time were greater and, in most cases, the animals succumbed earlier. Mice that did not receive antivenom died within the first two hours after inoculation, and lethality in the remaining groups was dose-dependent, with animals surviving significantly shorter time periods as venom/antivenom ratios increased, as expected. Although in the absence of antivenom *B. asper* killed animals 0.9 h faster than *L. stenophrys*, the drop in temperature was similar in animals exposed to one or the other venom, either mixed or not with antivenom. Based on the complexity of the venoms and their diversity among different species and subspecies of snakes, a difference in the severity of their effects is expected.

When we compared the animals that survived to those that did not, we showed that a drop in temperature of 6 °C predisposed animals to death, regardless of the venom type evaluated. This was evident within the first 3 h after treatment in all groups treated with antivenom and was independent of the baseline temperature and venom type. Importantly, regression analyses showed that with each 0.94° drop in temperature (relative to baseline), mice were 2.5 times more likely to die. As expected, increases in the venom/antivenom ratio also significantly increased the likelihood of death. Our results suggest that body temperature could be considered as an objective parameter to define the humane endpoint for neutralization-potency tests of antivenoms. This would considerably reduce the time of animals’ suffering. However, this parameter is specific to each experimental set up, so it must be properly validated for other venoms.

The WHO guidelines recommend evaluating lethality for a period of 48 h, after intraperitoneal administration [9]. Some studies have suggested that the lethality assessment can be reduced to 24 h [24] or even to 8 h [25]. Our results showed a lethality of 40.7% at 24 h and 53.9% at 48 h, indicating that the number of deaths could be underestimated if the observation time is reduced. This underestimation of mortality may affect the resulting ED_50_, and therefore, it may be unacceptable for the purpose of the test. However, at 48 h, there is a greater probability that some deaths caused by phenomena other than the lack of effect of the antivenom occur and could be recorded incorrectly. Considering the drop in temperature relative to baseline as a predictive factor, it is possible to determine which animals will die within the first 3 h of exposure to the mixtures of *B. asper* or *L. stenophrys* venoms and antivenom. In fact, hypothetical values of ED_50_, calculated considering dead those animals that had a drop in temperature of at least 6 °C at 3 h, coincide with the real ED_50_ values, as calculated from the mortalities at 48 h. Previous studies have shown that animals who exhibited a drop in core body temperature of approximately 5 degrees, at early time points of snake venom exposure, were predisposed to die [18]. The authors predicted the mortality of Swiss mice inoculated with a Lethal Dose 50 (LD_50_) of venoms from three different types of rattlesnakes. Although the values of temperature and time to predict mortality were different for each venom, the core body temperature, measured by using a thermistor probe, was a predictor of death in each case [18].

Different methods to determine the body temperature of animals have been used, including implantable microchip transponder, infrared thermometers, and rectal or tympanic probes [14,26,27,28,29,30]. Body temperature has been shown to be a valid parameter to define humane endpoints regardless of the method used for its measurement [15,26]. In one model of endotoxemia, animal death was predicted via the temperature drop independent of whether the measurement was made with an infrared thermometer or an implanted transponder [26]. It is important to note that basal temperature measured with infrared thermometers varies depending on the site of the measurement and can be substantially different than the core temperature measured by using implanted devices or other methods [26,28]. We have observed an average basal temperature of approximately 32 °C, similar to that obtained in other studies [17,26,31]. However, body temperature in mice is systematically higher when measured at the base of the sternum (35–37 °C) [30,32,33]. Some authors have suggested that the body temperature measured in the sternum region more strongly correlates to the core temperature than that taken on other body regions [33]. Differences in body temperature measurements depending on the measurement site in humans have also been intensely discussed [34,35,36]. Thus, appropriate validation of this parameter for humane endpoint purposes is necessary because different endpoints have been reported for different experimental set-ups [16,27,37].

An infrared thermometer is a non-invasive instrument to measure temperature and does not induce animal discomfort; it is suitable for large-scale experiments and does not require surgery or technical expertise. Thus, the infrared thermometer is suitable for temperature measurements during neutralization-potency tests. Temperature measurements with infrared thermometers are subjected to the device and user dependency; therefore, it is recommended to use the same thermometer by a single user in all the measurements made during the experimental procedures. This method partially depends on the user, since care must be taken to ensure that the measurement is always in the same site to avoid false readings [30]. Temperature as a parameter to define a humane endpoint must be validated for each specific experimental condition and different mice strains.

It is important to note that ambient temperature can directly impact the surface temperature. Body surface temperature is influenced via heat exchange with the environment. Some skin features, such as the presence of hair and local vascularity, determine the temperature of each area in relation to the environment [38]. Thermoregulatory mechanisms are essential to compensate the variabilities in ambient temperature [39,40]. Meanwhile, the core body temperature remains stable even with high changes in ambient temperature. In mice, it has been observed that in specific animal facility conditions, core body temperature, measured over long time intervals, does not differ even at ambient temperatures as different as 22 °C and 30 °C [40]. In humans, unlike mice, variations in body temperature are dependent on several factors such as age, gender, physical activity, health conditions, and others [40,41]. It has been shown via infrared analysis that as body mass increases, the variation in surface temperature to regulate heat exchange with the environment becomes more important [42]. In small mammals, like mice, core temperature control is more dependent on metabolic heat production, and skin heat exchange with the environment is less critical in maintaining stable core temperature [42].

Importantly, a significant strength of our studies lies in the number of subjects across groups, which helps reduce experimental errors and increases the reliability of the findings. Notably, effect sizes across analyses were moderate to large, indicating that our results have practical significance. Another strength is that roughly half of the animals in these studies survived, resulting in a relatively balanced distribution between survived and deceased cases. This balance enhances the accuracy of death prediction estimates, positively contributing to the overall precision of the findings.

In sum, we show that a drop in temperature greater than 6 °C within the first 3 h of inoculation with a combination of *B. asper* or *L. stenophrys* venoms and antivenom is a predictor of mortality in BALB/c mice. The 3 h time point seems to be the most appropriate to select the animals that should be humanely euthanized to avoid unnecessary suffering. Animals that survive, along with temperature changes smaller than 6 °C, also suffer the clinical manifestations associated with the toxicity of the venom; however, they can fully recover within 48 h. Although the suffering of animals that survive cannot be prevented, using temperature as a criterion for establishing a humane endpoint will reduce suffering in approximately 50% of the animals, which, according to our data, will succumb within 48 h. Neutralization potency tests to determine ED_50_ of antivenoms, in parallel with tests using body temperature changes as an endpoint, will be important to validate the results presented herein. This approach will generate the relevant information needed to refine the use of animals during regular quality control analyses. The validation of temperature drops as a criterion for humane endpoints must be performed for different venom types, mice strains, and temperature measurement methods. It is important to avoid the premature termination of experiments, which may lead to incomplete data collection, and ultimately, to the use of more animals for new tests. To the best of our knowledge, this is the first study to suggest that the temperature drop in mice during venom-neutralizing potency tests could serve as an objective criterion for predicting lethality in mice.

## 4. Materials and Methods

### 4.1. Animals

Female BALB/c mice, weighing 20–22 g, were provided by the INDICASAT Animal Facility. Animals were maintained with a 12 h light/dark cycle, at a constant temperature of 24 °C with free access to food and water. For the experiments, mice were group-housed at five mice per cage. All experimental procedures were approved by the Institutional Animal Care and Use Committee of INDICASAT (IACUC-22-005) and were based on the strict observance of the ethical guidelines related to the handling of laboratory animals, in accordance with international regulations and those established by INDICASAT.

### 4.2. Venom and Antivenom

The venoms of *Bothrops asper* and *Lachesis stenophrys* and polyvalent antivenom were produced by the Instituto Clodomiro Picado in Costa Rica and received through the Instituto Especializado de Análisis at the University of Panama. Venoms were received in the form of a lyophilized powder, stored at −20 °C and diluted in Phosphate-Buffered Saline (PBS) prior to use. Antivenoms were received as a liquid or lyophilized formulation that was preserved at 4 °C until use.

### 4.3. Venom-Neutralizing Potency Test

Mixtures of venom and anti-venom were prepared by adding a constant amount of venom (corresponding to 4xLD_50_ for each venom per animal) with variable concentrations of antivenom. The tested proportions were those corresponding to several hypothetical neutralization potencies, as experimentally designed to find the ED_50_ of the antivenom. These proportions were 2, 3, 4.5, and 6.75 mg of venom/mL of antivenom. Mixtures were incubated in a bath at 37 °C for 30 min. A total of five animals per group (four groups) were inoculated intraperitoneally (ip) with 0.5 mL of the mixtures. An additional control group inoculated only with venom resuspended in PBS was included. Death was recorded for a period of 48 h, as requested by the WHO protocols for anti-venom evaluations [9]. Median Effective Dose (ED_50_) was calculated via the Trimmed Spearman–Karber Method [23].

### 4.4. Temperature Measurement

A non-contact infrared thermometer (Lasergrip 774 Infrared Thermometer, Etekcity, Anaheim, CA, USA) was used to measure the surface temperature of animals in the perianal region. To measure the temperature, animals were held by the base of the tail to expose the anus and lifted slightly so that their front paws were supported on the metal rod of the cage lid. The infrared light of the thermometer was focused on the reading point at 1 cm for approximately 5 s. The temperature was measured just before inoculation (baseline), 1, 2, and 3 h post-inoculation and additionally, for the surviving animals, at 24 and 48 h.

### 4.5. Statistical Analysis

Descriptive analyses examined the average temperature across venom types (*L. stenophrys* vs. *B. asper*), dosages (2, 3, 4.5, and 6.75 mg venom/mL antivenom, and control groups). A two-way analysis of variance (ANOVA) examined the effect of the venom type and dosage on survival time. A hierarchical binary logistic regression was used to predict mortality from venom type and temperature change (from baseline to last recorded temperature), controlling for the baseline temperature. Lastly, repeated measures ANOVA was used to examine the temperature at baseline (0 h), 1 h, 2 h, and 3 h for animals that were exposed to venom and antivenom and had complete data at those time points to assess the differences between animals that survived and those that did not. A two-way analysis of variance (ANOVA) with Šidák’s multiple comparison test was used for the analysis of independent experiments. Significance was set at *p* < 0.05 and corrections were made for multiple comparisons. Analyses were conducted with SPSS version 28 and Prism GraphPad 9.

## Figures and Tables

**Figure 1 toxins-15-00525-f001:**
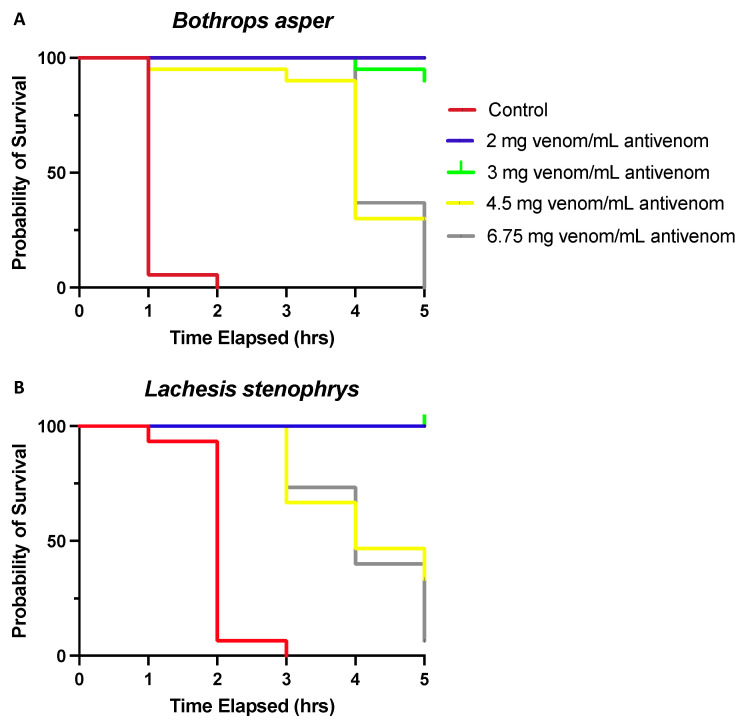
Kaplan–Meier survival analysis of (**A**) *B. asper* and (**B**) *L. stenophrys* venoms stratified into five groups (controls and 2, 3, 4.5, and 6.75 mg venom/mL antivenom) shows a significantly better overall survival for groups receiving lower venom/antivenom dosages. Lethality rates were not different between venoms.

**Figure 2 toxins-15-00525-f002:**
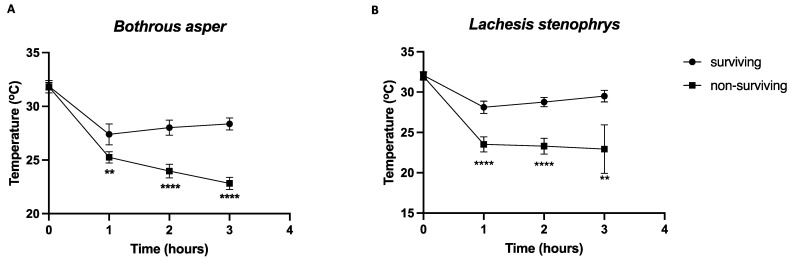
Temperature differences between surviving and non-surviving mice. Groups of five mice were inoculated with mixtures of a fixed dose of *B. asper* (**A**) or *L. stenophrys* (**B**) venoms (4xLD_50_) and variable doses of antivenom (2, 3, 4.5, and 6.75 mg venom/mL antivenom). Body temperature was recorded before inoculation and every hour for 3 h. Graphs represent mean ± 95% CI of temperature from surviving (*n* = 40 for *B. asper*; *n* = 39 for *L. stenophrys*) and non-surviving (*n* = 35 for *B. asper*; *n* = 35 for *L. stenophrys*) mice. Animals that did not survive had significantly lower temperatures at all time points (except at 0 h) relative to animals that survived, (** *p* < 0.01; **** *p* < 0.0001).

**Table 1 toxins-15-00525-t001:** Descriptive statistics for survival duration by venom type and group (dosage).

Venom Type	Venom mg/mL Antivenom Dosage	Mean *	Standard Deviation	*n*
*Bothrops asper*	2 mg of venom/mL antivenom	5.67	1.29	15
	3 mg venom/mL antivenom	5.80	0.56	15
	4.5 mg venom/mL antivenom	4.80	0.94	15
	6.75 mg venom/mL antivenom	4.40	0.51	15
	Control (venom only)	1.07	0.27	14
*Lachesis stenophrys*	2 mg of venom/mL antivenom	5.93	0.26	15
	3 mg venom/mL antivenom	6.00	0.00	15
	4.5 mg venom/mL antivenom	4.47	1.30	15
	6.75 mg venom/mL antivenom	4.07	0.83	14
	Group 5: Control (venom only)	2.00	0.38	15

* Note: Survival duration was coded as an interval variable: 1 = survived up until 1 h, 2 = survived up until 2 h, 3 = survived up until 3 h, 4 = survived up until 24 h, 5 = survived up until 48 h, and 6 = survived the whole study.

**Table 2 toxins-15-00525-t002:** Hierarchical binary logistic regression: Unstandardized regression coefficients, standard errors, and confidence intervals predicting non-survival.

Predictor Variables	*b*	Standard Error	Odds Ratio	95% CI
**Step 1**				
Baseline temperature	−0.08	0.15	0.92	0.69, 1.23
**Step 2**				
Baseline temperature	−0.11	0.15	0.90	0.67, 1.21
Venom type	0.34	0.36	1.41	0.70, 2.84
**Step 3**				
Baseline temperature	−0.58 *	0.26	0.56	0.33, 0.93
Venom type	0.85	0.59	2.33	0.73, 7.45
Temperature change	0.94 ***	0.16	2.58	1.89, 3.46
**Step 4**				
Baseline temperature	−0.82 *	0.40	0.44	0.20, 0.97
Venom type	0.98	0.88	2.67	0.47, 15.01
Temperature change	0.70 ***	0.20	2.02	1.37, 2.98
Venom mg/mL antivenom	2.78 ***	0.68	16.17	4.30, 60.85

Note: *p* < 0.05 * and *p* < 0.001 ***, CI = confidence interval.

**Table 3 toxins-15-00525-t003:** Change in temperature (relative to baseline) as a function of survival at 1-, 2-, and 3 h.

	Time of Temperature Measurement	Mean Temperature Change (*SD*)	95% CI
Outcome			
Survived	1 h	3.87 (0.38)	3.11, 4.63
	2 h	3.26 (0.32)	2.62, 3.90
	3 h	2.95 (0.30)	2.35, 3.54
Did not survive			
	1 h	6.88 (0.47)	5.95, 7.80
	2 h	7.91 (0.39)	7.13, 8.69
	3 h	8.88 (0.36)	8.15, 9.60

## Data Availability

The data presented in this study are available in the article and Appendix A.

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
