# Peer review of "Body Temperature Drop as a Humane Endpoint in Snake Venom-Lethality Neutralization Tests"

_toxins, 2023, doi:10.3390/toxins15090525_

Round 1

Reviewer 1 Report

The authors studied a drop in the body temperature of mice as a potential predictor of death induced by the venom inoculation. The research was driven by the idea of implementing the concept into the lethal-toxicity neutralization assay, as it might reduce the period during which the animals are exposed to pain and distress. While the paper addresses an interesting issue, I do not believe that Toxins is a good fit for it. The major criticism goes to the presentation of the collected data. In summary, the mixture of venom was preincubated with different concentrations of specific antivenom and inoculated into groups of five mice. For each venom/antivenom combination at defined time points a drop in the body temperature was measured and deaths were recorded. The authors could calculate ED50 value by the procedure recommended by WHO and compare it with the one determined by their method since, according to the presented work, based on the temperature change it is possible to predict which mice will survive at time points as early as 1 h from inoculation. But, the authors missed to validate the employed principle and just defined which drop in temperature predisposes animals to death. They could easily come to the same conclusion by testing only different concentrations of venom (without antivenom), but such investigation has already been performed by Charles et al., 2014. The following bothers me as well. It seems that several independent experiments were performed, but for the unknown reason all values were given cumulatively. Correct way would be to provide the mean value of each individual assay as the basis for determining the correlation between the temperature change and the mortality rate. I have too many concerns about the paper. To me, scientific significance of the investigation is poor and I do not recommend it for publication in Toxins.

Author Response

Panama July 31, 2023

Dear Reviewer,

We thank reviewer for the thorough review of the manuscript, and we hope the revised manuscript addresses your concerns.

All authors approved all changes in the manuscript.

Following are the comments and our responses:

Reviewer

The authors studied a drop in the body temperature of mice as a potential predictor of death induced by the venom inoculation. The research was driven by the idea of implementing the concept into the lethal-toxicity neutralization assay, as it might reduce the period during which the animals are exposed to pain and distress. While the paper addresses an interesting issue, I do not believe that Toxins is a good fit for it.

Reviewer comment: The major criticism goes to the presentation of the collected data. In summary, the mixture of venom was preincubated with different concentrations of specific antivenom and inoculated into groups of five mice. For each venom/antivenom combination at defined time points a drop in the body temperature was measured and deaths were recorded. The authors could calculate ED50 value by the procedure recommended by WHO and compare it with the one determined by their method since, according to the presented work, based on the temperature change it is possible to predict which mice will survive at time points as early as 1 h from inoculation. But, the authors missed to validate the employed principle and just defined which drop in temperature predisposes animals to death. They could easily come to the same conclusion by testing only different concentrations of venom (without antivenom), but such investigation has already been performed by Charles et al., 2014.

Authors response: We thank the reviewer for the comment. The experiments were carried out in the context of the quality control evaluation of antivenoms requested by the regulatory agency of Panama. For quality control evaluations were performed using venom-neutralizing potency tests and ED50 values were reported. The authors agree with the reviewer that LD50 experiments such as those reported by Charles et al., 2014 would also have been important in adding information to these data; however, the purpose of this work is to draw the attention of the scientific community to the importance of establishing objective parameters that allow refining the use of animals during quality control evaluations required by national regulatory agencies. This would reduce animal suffering during these kinds of experiments. For that reason, the temperature evaluation was carried out in lethality neutralization experiments and not in LD50 experiments. On the other hand, the data published by Charles et al., 2014 were using three rattlesnake venoms and the data in this manuscript were obtained by testing the venoms of Bothrous asper and Lachesis stenophrys. Considering the differences in venom composition for different snake species and subspecies, and the clinical manifestations induced by different species, the authors considered that it is important to report results obtained from other venom types. The authors agree that much more data will be required to be published in order to get a change in the official guidelines for these quality control assays regarding the use of body temperature replacing mortality.

ED50 data calculated according to WHO recommended procedures were included in supplementary material (Table S3). The authors agree with the reviewer that experiments calculating ED50 using temperature drop as a humane endpoint criterion are needed to validate the use of this parameter as a humane endpoint. To the best of our knowledge, this is the first study reporting temperature drop in mice during venom-neutralizing potency tests. The purpose of this first manuscript was to report the data related to the correlation between the drop in temperature at very early times after the inoculation of the venom/antivenom mixtures and the death of the animals. Validation experiments using this objective parameter as endpoint are the purpose of a further study. However, we calculated hypothetical ED50 values for each of the independent experiments at the 1-, 2-, and 3-hour time points, selecting those animals that had a drop in temperature of at least 6°C as those that would be euthanized. The 3-hour time point values were the most similar throughout the experiments to the real ED50 values calculated by evaluating the 48-hour mortality. These data are presented in the Supplementary Table S2 and discussed in the text of the revised manuscript.

Reviewer comment: The following bothers me as well. It seems that several independent experiments were performed, but for the unknown reason all values were given cumulatively. Correct way would be to provide the mean value of each individual assay as the basis for determining the correlation between the temperature change and the mortality rate. I have too many concerns about the paper.

Author response: We thank the reviewer for the comment. Data were presented cumulatively in order to achieve a greater statistical power. The independent experiments were performed with a small number of animals, since following the reduction principle, we used the lowest number of animals per group recommended by the WHO for this kind of experiment (5 mice/group). Three independent experiments were performed for each venom. In the revised manuscript, Figure 2, we included two graphs that correspond to the cumulative data from the 3 experiments for each venom. Results per experiment were included as supplementary material (Figure S1). Some of the experiments for L. stenophrys were missing temperature measurements at 3 hours; however, considering that at 1 hour we were able to find a significant difference between surviving and non-surviving mice, the data from those experiments calculated only for 1- and 2-hours post-inoculation were also presented.

Respectfully,

Reviewer 2 Report

The paper proposes an objective parameter, such as temperature measurement, to establish a humane endpoint in venom neutralization assays.

According to the authors, it is possible to predict which animals will survive after the inoculation of venom and antivenom by measuring the variation in the animals’ body temperature in relation to the baseline, avoiding prolonged suffering of the animals.

The drop in body temperature is correlated with death in animal models of infectious diseases, and has been demonstrated in animals exposed to snake venom.

In this paper, the authors demonstrated that depending on the value of the temperature drop, it is possible to predict the death of the animal as early as the first hour after inoculation.

It is a valid suggestion for the reduction in the number of animals in the tests evaluating the effectiveness of the antivenom that must be taken into account.

Author Response

Panama July 31, 2023

Dear Reviewer,

We thank reviewer for the thorough review of the manuscript, and we hope the revised manuscript addresses your concerns.

All authors approved all changes in the manuscript.

Following are the comments and our responses:

Reviewer 

The paper proposes an objective parameter, such as temperature measurement, to establish a humane endpoint in venom neutralization assays.

According to the authors, it is possible to predict which animals will survive after the inoculation of venom and antivenom by measuring the variation in the animals’ body temperature in relation to the baseline, avoiding prolonged suffering of the animals.

The drop in body temperature is correlated with death in animal models of infectious diseases, and has been demonstrated in animals exposed to snake venom.

In this paper, the authors demonstrated that depending on the value of the temperature drop, it is possible to predict the death of the animal as early as the first hour after inoculation.

It is a valid suggestion for the reduction in the number of animals in the tests evaluating the effectiveness of the antivenom that must be taken into account.

Authors response: We thank the reviewer for the positive comments on the manuscript.

Respectfully,

Reviewer 3 Report

The manuscript describes results showing that the change in body temperature of mice can be used as a criterion for predicting early death. This, in turn, could be used in the quality control of antivenoms, to replace the currently applied protocol. Assays are performed using venoms from two species of snakes and their antivenoms. The results, although simple, are explored through extensive statistical analysis, which makes the data more robust. My analysis is that the manuscript is more relevant for its applicability than for the data it adds.

Major comment:

The way the results are presented and discussed leaves a perception that the use of temperature change from baseline as a criterion to perform quality control analyses of antivenoms may apply to all snake venoms. As only two poisons were tested, I believe this is not possible. Therefore, I think it is fundamental that this is made clear to the reader. It might be interesting to insert these ideas after the sentence in line 179.

Minnor comments:

11. Results described in item 2.2 should be discussed in the “Discussion” section. This, besides enriching the discussion, could highlight the analyzes carried out. As it stands, these analyzes have not been properly valued.

2. In general, the results were these, as described by the authors:

- Mice exposed to the toxic effects of venom from B. asper or L. stenophrys showed a sudden drop in body temperature

- This decrease in temperature became more marked in experimental groups where the ratio venom/antivenom increased

- Mice that did not receive antivenom died within the first two hours after inoculation.

- Although in the absence of antivenom B. asper killed animals 0.9 hr faster than L. stenophrys, the drop in temperature was similar in animals exposed to one or the other venom, either mixed or not with antivenom.

- A drop in temperature of 6 degrees predisposed animals to death, regardless of venom type. This was evident within 1 hour after treatment in all groups treated with antivenom and was independent of baseline temperature.

However, the manuscript presents four tables and two figures. The results are shown in different ways, because each analysis highlights a different aspect. But these aspects are not addressed in the discussion, showing a lack of relevance of such analyzes. In this way, it is essential that the discussion is substantially improved.

Author Response

Panama July 31, 2023

Dear Reviewer,

We thank reviewer for the thorough review of the manuscript, and we hope the revised manuscript addresses your concerns.

All authors approved all changes in the manuscript.

Following are the comments and our responses:

Reviewer

The manuscript describes results showing that the change in body temperature of mice can be used as a criterion for predicting early death. This, in turn, could be used in the quality control of antivenoms, to replace the currently applied protocol. Assays are performed using venoms from two species of snakes and their antivenoms. The results, although simple, are explored through extensive statistical analysis, which makes the data more robust. My analysis is that the manuscript is more relevant for its applicability than for the data it adds.

Major comment:

The way the results are presented and discussed leaves a perception that the use of temperature change from baseline as a criterion to perform quality control analyses of antivenoms may apply to all snake venoms. As only two poisons were tested, I believe this is not possible. Therefore, I think it is fundamental that this is made clear to the reader. It might be interesting to insert these ideas after the sentence in line 179.

Authors response: We thank the reviewer for the comment. As recommended by the reviewer, we include throughout the sections of the revised manuscript clarifying statements that the data is specific for the venoms of B. asper and L. stenophrys.

Minnor comments:

  1. Results described in item 2.2 should be discussed in the “Discussion” section. This, besides enriching the discussion, could highlight the analyzes carried out. As it stands, these analyzes have not been properly valued.

Authors response: We thank the reviewer for the comment. The statistical analyses were fully explained in the results and as recommended by the reviewer we have expanded the relevance of these analyses in the discussion.

  1. In general, the results were these, as described by the authors:

- Mice exposed to the toxic effects of venom from B. asper or L. stenophrys showed a sudden drop in body temperature

- This decrease in temperature became more marked in experimental groups where the ratio venom/antivenom increased

- Mice that did not receive antivenom died within the first two hours after inoculation.

- Although in the absence of antivenom B. asper killed animals 0.9 hr faster than L. stenophrys, the drop in temperature was similar in animals exposed to one or the other venom, either mixed or not with antivenom.

- A drop in temperature of 6 degrees predisposed animals to death, regardless of venom type. This was evident within 1 hour after treatment in all groups treated with antivenom and was independent of baseline temperature.

However, the manuscript presents four tables and two figures. The results are shown in different ways, because each analysis highlights a different aspect. But these aspects are not addressed in the discussion, showing a lack of relevance of such analyzes. In this way, it is essential that the discussion is substantially improved.

Authors response: We thank the reviewer for the comment. Following the reviewers' recommendations in the revised manuscript, the presentation of some of the data has been modified. In addition, we have moved data into the supplementary material and have included new graphs and tables in the supplementary material. The results and discussion sections have been expanded. We consider that these changes have enriched the manuscript and have highlighted the importance of the results presented.

Respectfully,

Reviewer 4 Report

Major comments:

1. the results obtained by the authors should be validated using standard ED50 assay and its principle of the result's interpretation - until this is done, the appropriateness of the method in the quality control is questionable

2. the results of each experiment should be analyzed separately, not cumulatively 

3. it should be precisely described how the lethal toxicity neutralization assay was performed and how the potency of the antivenom was calculated (I think that Spearmen-Karber method is the recommended one)

I am not willing to continue participating in the review process furtherly until those issues are resolved. 

Author Response

Panama July 31, 2023

Dear Reviewer,

We thank reviewer for the thorough review of the manuscript, and we hope the revised manuscript addresses your concerns.

All authors approved all changes in the manuscript.

Following are the comments and our responses:

Reviewer

Major comments:

Reviewer comment: The results obtained by the authors should be validated using standard ED50 assay and its principle of the result's interpretation - until this is done, the appropriateness of the method in the quality control is questionable

Authors response: We thank the reviewer for the comment. The authors agree with the reviewer that experiments calculating ED50 using temperature drop as a humane endpoint criterion are needed to validate the use of this parameter as a humane endpoint. To the best of our knowledge, this is the first study reporting temperature drop in mice during lethality neutralization tests. The purpose of this first manuscript was to report the data related to the correlation between the drop in temperature at very early time points after the inoculation of the venom/antivenom mixtures and the death of the animals. Validation experiments using this objective parameter as endpoint are the purpose of a further study. However, we calculated hypothetical ED50 values for each of the independent experiments at the 1-, 2-, and 3-hour time points, selecting those animals that had a drop in temperature of at least 6°C as those that would be euthanized. The 3-hour time point values were the most similar throughout the experiments to the real ED50 values calculated by evaluating the 48-hour mortality. These data are presented in the Supplementary material (Tables S2 and S3) and discussed in the text of the revised manuscript.

Reviewer comment: The results of each experiment should be analyzed separately, not cumulatively 

Authors response: We thank the reviewer for the comment. Data were presented cumulatively in order to achieve a greater statistical power. The independent experiments were performed with a small number of animals, since following the reduction principle, we used the lowest number of animals per group recommended by the WHO (5 mice/group) for this kind of experiment. Three independent experiments were performed for each venom. In the revised manuscript, Figure 2, we included two graphs that correspond to the cumulative data from the 3 experiments for each venom. Results per experiment were included as supplementary material (Figure S1). Some of the experiments were missing temperature measurements at 3 hours; however, considering that at 1 hour we were able to find a significant difference between surviving and non-surviving mice, the data from those experiments calculated only for 1- and 2-hours post-inoculation were also presented.

Reviewer comment:  it should be precisely described how the lethal toxicity neutralization assay was performed and how the potency of the antivenom was calculated (I think that Spearmen-Karber method is the recommended one)

Authors response: We thank the reviewer for the comment. As requested by the reviewer, we included in the item Venom-neutralizing potency test (Materials and methods section) how ED50 was calculated (Lines 730-731 from the revised version of the manuscript).

Respectfully,

Reviewer 5 Report

The manuscript “Body temperature drop as a humane endpoint in antivenom lethality neutralization tests” describes a new method for measuring the potency of antivenoms.

This approach is original and very interesting. Initial results seem promising, relevant, faster than the current method and, above all, more ethical with regard to laboratory animals.

The methodology is appropriate and reliable. The results are well presented. The discussion could be significantly improved by moving towards a broader validation of the method, and by preparing more precise advocacy for its promotion.

Indeed, before replacing the tests approved and recommended by the WHO, it is necessary to check their accuracy for the various venoms and antivenoms, to standardize the method, instrument and implementation, and then to argue to convince the stakeholders.

Major concerns

The discussion should more clearly address the following issues.

1. The most critical is to replace a binary outcome (dead/alive, regardless of the condition of the animal) by a continuous variable which is moreover, a substitution endpoint. In the latter case, the threshold is difficult to determine and very often remains irrelevant to obtain the result of the test. This defect would be added to those that we already know, such as the questionable representativeness of mice compared to humans to express sensitivity to the effects of venom and protection by antivenom.

2. The proposed method must be assessed for other venoms, in particular those of elapids which are generally less inflammatory and could determine less marked temperature variations. However, there are many polyvalent antivenoms against vipers and elapids, which must be validated using the same experiment.

3. It would be useful to provide a general guide to achieve the standardization of this method.

Specific comments

Lines 188-192. I agree with the authors. However, the choice of 24 hours has 2 advantages: a) it allows standardization of the method and b) it is a fair compromise limiting the risk of wrongly including deaths that would be due to an indirect cause (sepsis, for example), or even an independent cause, not directly linked to the absence of neutralization of the venom by the antivenom.

Lines 208-211. The authors do not mention variations in body surface temperature depending on where it is measured. This point was debated during the Covid-19 pandemic (for mass measurement of body temperature) and led to specific studies (see Chan et al., 2022; doi: 10.3390/ijerph192315883; Lippi et al. 2021. doi: 10.1515/dx-2021-0091).

Lines 223-225. I don't understand how multiple people's temperature readings can affect the variability of body temperature according to ambient temperature. The latter causes the body temperature to vary more or less markedly depending on other parameters (age, body mass, health condition, possibly the location of the measurement or the type of thermometer) but not on the person performing the reading. On the other hand, the variations could be proportional to the severity of the envenomation (see on the topic of temperature variations according to health condition, Stéphan et al. 2005; doi: 10.1093/bja/aeh291). This issue deserves a more specific discussion.

Minor comments

Authors of reference 21 need to be checked and corrected.

Author Response

Panama July 31, 2023

Dear Reviewer,

We thank reviewer for the thorough review of the manuscript, and we hope the revised manuscript addresses your concerns.

All authors approved all changes in the manuscript.

Following are the comments and our responses:

Reviewer

The manuscript “Body temperature drop as a humane endpoint in antivenom lethality neutralization tests” describes a new method for measuring the potency of antivenoms.

This approach is original and very interesting. Initial results seem promising, relevant, faster than the current method and, above all, more ethical with regard to laboratory animals.

The methodology is appropriate and reliable. The results are well presented. The discussion could be significantly improved by moving towards a broader validation of the method, and by preparing more precise advocacy for its promotion.

Indeed, before replacing the tests approved and recommended by the WHO, it is necessary to check their accuracy for the various venoms and antivenoms, to standardize the method, instrument and implementation, and then to argue to convince the stakeholders.

Major concerns

The discussion should more clearly address the following issues.

Reviewer comment:  The most critical is to replace a binary outcome (dead/alive, regardless of the condition of the animal) by a continuous variable which is moreover, a substitution endpoint. In the latter case, the threshold is difficult to determine and very often remains irrelevant to obtain the result of the test. This defect would be added to those that we already know, such as the questionable representativeness of mice compared to humans to express sensitivity to the effects of venom and protection by antivenom.

Authors response: We thank the reviewer for the comment. We agree with the reviewer that a threshold is difficult to establish for a continuous variable. However, we calculate the average temperature drop relative to baseline for animals that did not survive, this was 6.9°C (95% CI: 6.0-7.8). Then, we considered that animals that reach a drop in temperature of at least 6°C, which corresponds to the lower value of the confidence interval, will not survive. We calculated hypothetical ED50 values for each of the independent experiments at the 1-, 2-, and 3-hour time points, selecting those animals that had a drop in temperature of at least 6°C as those that would be euthanized. For both venoms, most of the values for all time points were similar to the ED50 calculated based on mortality at 48 hours. However, the 3-hour time point values were the most similar throughout the three experiments to the real ED50 values. So, we consider that this time point appears to be the most appropriate to select the animals that should be euthanized to avoid suffering. These data are presented in the Supplementary Table S2 and discussed in the text of the revised manuscript. However, we point out that a validation of this parameter as a humane endpoint needs further studies.

Reviewer comment:  The proposed method must be assessed for other venoms, in particular those of elapids which are generally less inflammatory and could determine less marked temperature variations. However, there are many polyvalent antivenoms against vipers and elapids, which must be validated using the same experiment.

Authors response: We thank the reviewer for the comment. We agree with the reviewer. However, the data showed in the present manuscript were obtained in the context of antivenom quality control analysis as part of the requirements of the Panamanian regulatory agency. We are currently evaluating antivenom to other venoms and collecting data on another strain of mice. These data will be the subject of another manuscript that will enrich the discussion in the scientific community on this matter. Anyway, the use of temperature measurement as a criterion for humane endpoint must be adequately validated by each quality control laboratory according to the specific experimental conditions (type of venom, mouse strain, temperature measurement method).

The purpose of this manuscript is not to describe a new method to calculate the potency of snake antivenoms, but rather to call attention to taking temperature drop into account as a potential humane endpoint to reduce animal suffering. This would necessarily lead to the further validation of this parameter to use it as a criterion in lethal neutralization analyses.

Reviewer comment:  It would be useful to provide a general guide to achieve the standardization of this method.

Authors response: We thank the reviewer for the comment. We consider that the preparation of guidelines for the standardization of temperature drop as a humane endpoint criterion is outside the scope of this manuscript. The purpose of this manuscript was to report the data related to the correlation between the drop in temperature at very early time points after the inoculation of the B. asper or L. stenophrys venom/antivenom mixtures and the death of the animals. To the best of our knowledge, this is the first study to report temperature drop in mice during venom-lethality neutralization tests. Further experiments with other venoms and validation of the use of temperature measurement as endpoint criteria will be needed to drive the preparation of guidelines by competent organizations.

Specific comments

Reviewer comment:  Lines 188-192. I agree with the authors. However, the choice of 24 hours has 2 advantages: a) it allows standardization of the method and b) it is a fair compromise limiting the risk of wrongly including deaths that would be due to an indirect cause (sepsis, for example), or even an independent cause, not directly linked to the absence of neutralization of the venom by the antivenom.

Authors response: We thank the reviewer for the comment. We agree with the reviewer. As recommended by the reviewer, we have added a statement in the discussion referring to this point (lines 407-409 from the revised manuscript).

Reviewer comment:  Lines 208-211. The authors do not mention variations in body surface temperature depending on where it is measured. This point was debated during the Covid-19 pandemic (for mass measurement of body temperature) and led to specific studies (see Chan et al., 2022; doi: 10.3390/ijerph192315883; Lippi et al. 2021. doi: 10.1515/dx-2021-0091).

Authors response: We thank the reviewer for the comment. As suggested by the reviewer we have expanded the discussion on this specific topic (lines 466-477 from the revised manuscript).

Reviewer comment:  Lines 223-225. I don't understand how multiple people's temperature readings can affect the variability of body temperature according to ambient temperature. The latter causes the body temperature to vary more or less markedly depending on other parameters (age, body mass, health condition, possibly the location of the measurement or the type of thermometer) but not on the person performing the reading. On the other hand, the variations could be proportional to the severity of the envenomation (see on the topic of temperature variations according to health condition, Stéphan et al. 2005; doi: 10.1093/bja/aeh291). This issue deserves a more specific discussion.

Authors response: We thank the reviewer for the comment. We agree with the reviewer. As suggested by the reviewer we have expanded the discussion on this topic (lines 488-502 from the revised manuscript).

Minor comments

Reviewer comment:  Authors of reference 21 need to be checked and corrected.

Authors response: We have corrected the reference 21.

Respectfully,

Round 2

Reviewer 1 Report

No more comments.

Reviewer 3 Report

My suggestions were accepted, and the changes made. Important changes were made beyond my suggestions. I consider that in this way the manuscript is suitable for publication.

Reviewer 5 Report

The authors responded satisfactorily to the remarks and recommendations.